# Hyaluronic Acid as Macromolecular Crowder in Equine Adipose-Derived Stem Cell Cultures

**DOI:** 10.3390/cells10040859

**Published:** 2021-04-09

**Authors:** Sergio Garnica-Galvez, Stefanie H. Korntner, Ioannis Skoufos, Athina Tzora, Nikolaos Diakakis, Nikitas Prassinos, Dimitrios I. Zeugolis

**Affiliations:** 1Laboratory of Animal Science, Nutrition and Biotechnology, Department of Agriculture, University of Ioannina, 47100 Arta, Greece; gsergio@vet.auth.gr (S.G.-G.); jskoufos@uoi.gr (I.S.); tzora@uoi.gr (A.T.); 2School of Veterinary Medicine, Aristotle University of Thessaloniki, 54124 Thessaloniki, Greece; diakakis@vet.auth.gr (N.D.); ngreen@vet.auth.gr (N.P.); 3Regenerative, Modular & Developmental Engineering Laboratory (REMODEL), Biomedical Sciences Building, National University of Ireland Galway (NUI Galway), H92 W2TY Galway, Ireland; stefanie.korntner@gmail.com; 4Science Foundation Ireland (SFI) Centre for Research in Medical Devices (CÚRAM), Biomedical Sciences Building, National University of Ireland Galway (NUI Galway), H92 W2TY Galway, Ireland; 5Regenerative, Modular & Developmental Engineering Laboratory (REMODEL), Faculty of Biomedical Sciences, Università della Svizzera Italiana (USI), 6904 Lugano, Switzerland; 6Regenerative, Modular & Developmental Engineering Laboratory (REMODEL), School of Mechanical and Materials Engineering, University College Dublin (UCD), D04 V1W8 Dublin, Ireland

**Keywords:** excluded volume effect, extracellular matrix deposition, organogenesis

## Abstract

The use of macromolecular crowding in the development of extracellular matrix-rich cell-assembled tissue equivalents is continuously gaining pace in regenerative engineering. Despite the significant advancements in the field, the optimal macromolecular crowder still remains elusive. Herein, the physicochemical properties of different concentrations of different molecular weights hyaluronic acid (HA) and their influence on equine adipose-derived stem cell cultures were assessed. Within the different concentrations and molecular weight HAs, the 10 mg/mL 100 kDa and 500 kDa HAs exhibited the highest negative charge and hydrodynamic radius, and the 10 mg/mL 100 kDa HA exhibited the lowest polydispersity index and the highest % fraction volume occupancy. Although HA had the potential to act as a macromolecular crowding agent, it did not outperform carrageenan and Ficoll^®^, the most widely used macromolecular crowding molecules, in enhanced and accelerated collagen I, collagen III and collagen IV deposition.

## 1. Introduction

Cell-assembled tissue engineering has the potential to revolutionise reparative biomedicine, as it takes advantage of the cells’ natural intrinsic abilities to fabricate tissues with complexity, precision and efficiency that human-made devices have yet to reach. Despite the repeatedly demonstrated safety and efficacy in the clinical setting (e.g., cardiac [1], cartilage [2], cornea [3] and skin [4]), only a handful of concepts have passed the ‘valley of death’ to become commercial reality. It is accepted that the major culprit in the development of cell-based scaffold-free tissue engineering products is the lengthy period of time ex vivo (up to 196 days for blood vessels [5]) that is associated with losses in cell function [6] and extremely high manufacturing costs [7]. To this end, strategies that enhance and accelerate extracellular matrix (ECM) synthesis and deposition, whilst controlling cell fate during in vitro expansion should be incorporated into the developmental cycle of advanced therapy medicinal products to develop clinically and commercially relevant products.

It has been well-documented that among the various in vitro microenvironment modulators (e.g., surface topography [8], substrate rigidity [9], oxygen tension [10], mechanical stimulation [11] and growth factor supplementation [12]), macromolecular crowding (MMC) is the only one that profoundly enhances (up to 120-fold) and accelerates (within 6 days in culture) ECM deposition, due to its negative charge and high polydispersity [13,14,15]. This unprecedented success of MMC in eukaryotic cell culture lays on the fact that in traditionally used dilute cell culture systems, the enzymatic conversion of water-soluble procollagen to water-insoluble collagen is very slow [16], as the proteinases and the procollagen are diffused in the culture media, whilst in crowded cell culture systems, this enzymatic processing of collagen is rapid, as the proteinases and the procollagen are confined at the interface of cell-layer/crowders [17]. The effectiveness of MMC is based on its mechanism of action; due to steric hindrance and/or electrostatic repulsion, macromolecules in highly volume-occupied or crowded solutions exclude each other from their respective vicinity, as two macromolecules cannot occupy the same space at the same time (excluded volume effect) [18]. This excluded volume effect decreases diffusion in a system, via the obstruction of molecular motion [19], which unavoidably affects the thermodynamics, kinetics and equilibria of biochemical reactions and biological processes (e.g., DNA structure, stability, condensation and replication [20,21,22]; RNA and protein transcription, structure and folding [23,24,25]), subject to the physicochemical properties (e.g., charge [26], dispersity [27], size [28], shape [29] and viscosity [30]) of the MMC agents.

In cell culture, several macromolecules have been assessed as MMC agents over the years, largely categorised as non-sulphated polysaccharides (e.g., Ficoll^®^ [31]), sulphated polysaccharides (e.g., carrageenan [15]) and polymers (e.g., polyvinylpyrrolidone [32]). Although carrageenan, due to its negative charge and polydispersity, has been shown to induce the highest ECM deposition (up to a 120-fold increase) in the shortest period of time (within 6 days) [13], its questionable association with colitis [33,34,35] imposes the need to identify alternative molecules. In the quest for the ideal MMC agent, hyaluronic acid (HA) was recently utilised in human neonatal fibroblast cultures [36]. Although this proof of principle study demonstrated the potential of 1500 kDa HA to act as an MMC agent, the optimal HA molecular weight and concentration were not identified and the physicochemical properties (e.g., charge, hydrodynamic radius, polydispersity) were not assessed, which are of significant importance, considering that the effectiveness of HA as an MMC/volume exclusion agent lays on its capacity to bind and retain water molecules (up to 6 L per g) [37,38] in a size-, shape-, concentration- and viscosity-dependent manner [39]. Herein, we ventured to comprehensively assess the physicochemical properties of different concentrations of different molecular weight HAs in equine adipose-derived stem cell (eADSC) cultures.

## 2. Materials and Methods

### 2.1. Materials

Cell culture plasticware and chemicals/reagents were purchased from Sarstedt (Nümbrecht, Germany) and Sigma-Aldrich (Athens, Greece), respectively, unless otherwise stated (catalogue numbers are mentioned in brackets). HAs of different molecular weight (10 (10 to 20 kDa), 60 (66 to 99 kDa), 100 (100 to 150 kDa), 500 (301 to 450 kDa) and 1000 kDa (750 to 1000 kDa)) were purchased from Lifecore™ Biomedical (Chaska, MN, USA). Equine fat tissues, from the mane, were provided from the School of Veterinary Medicine, Aristotle University of Thessaloniki, Thessaloniki, Greece.

### 2.2. Stem Cell Extraction, Expansion, Cryopreservation and Thawing

Fat samples were washed with Hanks’ Balanced Salt Solution (HBSS, XC-S2064, Biosera, Nuaille, France) containing 1% penicillin/streptomycin (P/S, P4333), minced with sterile dissection material and enzymatically digested with 0.1% collagenase from *Clostridium histolyticum* (C9722). Collagenase digestion was performed for 4 h at 37 °C in standard cell culture media (Dulbecco’s Modified Eagle Medium high glucose (DMEM, D6429), 10% foetal bovine serum (FBS, F7524) and 1% P/S). Fat tissue stromal vascular fraction (SVF) was obtained by centrifugation of digested samples for 5 min at 700 g (4 times). SVF was resuspended in standard cell culture media and passed through a Corning^®^ nylon cell strainer of 70 μm (CLS431751). Nucleated cells from the SVF were counted using a Neubauer chamber and seeded at a density of 2,000,000 cells/cm^2^ in tissue culture flasks at 37 °C in a humidified atmosphere of 5% CO_2_. Media were changed every 3–4 days (the cells were washed twice with 1% P/S in phosphate buffered saline, PBS, to remove non-attached cells before fresh media were added) until the cells reached ~ 80% confluency. Cell detachment was carried out using trypsin-ethylenediaminetetraacetic acid (EDTA) solution (T3924) for 5 min at 37 °C in a humidified atmosphere of 5% CO_2_. Cell suspensions were centrifuged for 5 min at 700 g and then seeded at a density of 5000 cells/cm^2^ in standard cell culture media (DMEM, 10% FBS, 1% P/S) and conditions (37 °C, humidified atmosphere of 5% CO_2_). At passage 2–3, cell culture flasks were tested for mycoplasma [40]. Mycoplasma-free cells were stored in liquid nitrogen in 1.6 mL CryoPure tubes, each containing 1.8–2.0 × 10^6^ cells in 1 mL 90% FBS /10% dimethyl sulfoxide (DMSO). The cryovials were frozen at −80 °C in a Nalgene^®^ Mr. Frosty freezing container (C1562) and transferred to the liquid nitrogen storage after 24 h. Prior to use, cryovials were thawed in a thermostatic bath at 37 °C and cells were seeded in pre-equilibrated tissue culture flasks in standard cell culture media and conditions at 5000 cells/cm^2^. After 24 h, the media were changed to ensure complete removal of DMSO. From then onwards, the media were changed every 3–4 days until reaching 80% confluency and passage 4. All experiments were conducted with cells at passage 5.

### 2.3. Cell Morphology Analysis

Cell morphology was assessed via bright field microscopy using an inverted Olympus IX73 microscope (Olympus Corporation, Tokyo, Japan).

### 2.4. Stem Cell Characterisation Analysis

Multipotency was assessed with established osteogenic, adipogenic and chondrogenic protocols, using cells in DMEM as negative controls, and surface markers were assessed with fluorescence-activated cell sorting (FACS) analysis, validating the antibodies with immunocytochemistry.

#### 2.4.1. Osteogenic Analysis

For osteogenesis, 5000 cells/cm^2^ were seeded in 6-well plates. Standard cell culture media was substituted after 24 h by osteo-induction media consisting of 100 nM dexamethasone (D4902), 100 μM L-ascorbic acid 2-phosphate (A8960) and 10 mM β-glycerophosphate disodium salt hydrate (G9422) in 10% FBS and 1% P/S. Osteo-induction media were changed every 3 days up to day 14. At day 14, cell layers were treated with methanol at 4 °C for 20 min and calcium depots were detected by adding 400 μL of 40 mM alizarin red S stain (A5533), pH 4.1. Excess dye was removed with PBS washes.

#### 2.4.2. Adipogenic Analysis

For adipogenesis, 2500 cells/cm^2^ were seeded in 6-well plates. After 3 days the media were changed to adipo-induction media consisting of 0.1 μM dexamethasone, 500 μM 3-isobutyl-1-methylxanthine (IBMX, I5879), 200 μM indomethacin (I7378), 15% horse serum (H1270) and 1% P/S. The adipo-induction media was changed every 3 days up to day 18. At day 18, cell layers were treated with 10% formalin for 15 min at room temperature and lipid droplets were detected by adding 400 μL of 0.3% red oil O stain (O1391) in 60% isopropanol. Excess dye was removed with PBS washes.

#### 2.4.3. Chondrogenic Analysis

For chondrogenesis, pellet cultures were performed. Approximately 500,000 cells were pelleted into 15 mL conical polypropylene tubes with chondro-induction media consisting of serum-free DMEM High Glucose, 100 nM dexamethasone, 100× ITS + 1 (Insulin-transferrin-sodium selenite, linoleic-bovine serum albumin, BSA) liquid media supplement (I2521; 1× dilution in the final chondro-induction media), 40 µg/mL L-proline (P5607), 25 µg/mL L-ascorbic acid 2-phosphate, 10 ng/mL TGF-β3 (R&D Systems, UK, 8420-B3/CF) and 1% P/S. The chondro-induction media were changed every 3 days up to day 21. At day 21, pellets were cryoprotected with increasing gradient (15 and 30%) sucrose solutions. Cryoprotected pellets were embedded in optimal cutting temperature compound and frozen in liquid nitrogen. Sections 5 µm thick were obtained using a Leica CM1850 (Leica Microsystems, Milton Keynes, UK) cryotome. Sections were stained with Alcian blue (B8438), washed with 1% acetic acid, stained with nuclear fast red aluminium solution 0.1% (100121), washed with double-distilled water, dehydrated in 100% ethanol and mounted using DPX mountant for histology (06522). Images of osteogenic and adipogenic induction were captured with an Olympus IX73 microscope (Olympus Corporation, Tokyo, Japan) and images of chondrogenic induction were captured with an Olympus BX51 microscope (Olympus Corporation, Tokyo, Japan).

#### 2.4.4. FACS Analysis

A FACScalibur™ (BD, Wokingham, UK) device was used for cell surface markers analysis. Briefly, at 80% confluency at passage 5, cells were trypsinised and suspended at 2 × 10^6^ cells/mL density in 2 mM EDTA/PBS buffer. 50 μL aliquots were tested in flow cytometry tubes and incubated for 30 min at 4 °C with the following antibodies: AlexaFluor^®^ 488 anti-human CD29 (BioLegend^®^, San Diego. CA, USA, clone TS2/16) [41,42], FITC mouse anti-human CD34 (BD Pharmingen™, Wokingham, UK, clone 581) [43,44,45], FITC mouse anti-horse CD44 (Bio-Rad Laboratories, Watford, Hertfordshire, UK, clone CVS8) [43,44,45], PE mouse anti-human CD90 (BD Pharmingen™, Wokingham, UK, clone 5E10) [41,42,46] and PE mouse anti-human CD105 (ThermoFisher Scientific, Gloucester, UK, clone SN6) [44]. PE- and FITC-conjugated mouse IgG1, κ isotype controls (BD Pharmingen™, Wokingham, UK, clone MOPC21), were used to discard unspecific binding and cells alone as negative controls to eliminate autofluorescence. SYTOX™ Red Dead Cell Stain (ThermoFisher Scientific, Gloucester, UK, S3459) at a final concentration of 1 μM was used to distinguish viable and non-viable cells. Analysis was performed by using BD CellQuest™ software (BD, Wokingham, UK).

To validate the antibodies used for FACS analysis, immunocytochemistry was also carried out when the cells reached 80% confluency. Secondary antibodies were not used, as the CD markers used for FACS analysis were directly conjugated to the aforementioned fluorochromes. The antibodies were incubated at 4 °C overnight. Details of the immunocytochemistry analysis are provided below.

### 2.5. Solubility Analysis

In order to identify working concentrations of the different molecular weight (10, 60, 100, 500 and 1000 kDa) HAs, different concentrations (0.1, 0.5, 1, 5, 10, 20, 30, 50, 60, 100, 200 and 500 mg/mL) of each HA molecular weight were dissolved in standard cell culture media at 37 °C. Optimal concentrations from this experiment were also screened with cells in standard cell culture media under standard cell culture conditions after 4 days in culture (see below).

### 2.6. Dynamic Light Scattering Analysis

Zeta potential, polydispersity index and hydrodynamic radius were assessed using dynamic light scattering (Zetasizer ZS90, Malvern Instruments, Malvern, UK). The crowding solutions were prepared in PBS to mimic physiological conditions. Fractional volume occupancy was calculated using the obtained values of hydrodynamic radius for the different MMC agents, as has been described before [47].

### 2.7. Stem Cell Culture

Cells were seeded at a 15,000 cells/cm^2^ density in standard cell culture media supplemented with 100 μM L-ascorbic acid 2-phosphate to induce collagen synthesis. After 24 h, media were changed and replaced with media containing the different concentrations of the different HA molecular weights. Media without MMC and media with carrageenan (75 µg/mL, C1013) and Ficoll^®^ (cocktail of 37.5 mg/mL Ficoll^®^ 70 kDa, F2878 and 25 mg/mL Ficoll^®^ 400 kDa, F4375) were used as controls. All crowders were sterilised under UV light for 15 min and solubilised in standard cell culture media supplemented with L-ascorbic acid 2-phosphate at 37 °C until no particles in suspension were observed. Media changes were performed every 2 days until reaching day 8. The workflow is schematically illustrated in Appendix A.

### 2.8. Stem Cell Viability Analysis

A Live/Dead^®^ viability kit (L3224, Invitrogen™, Gloucester, UK) was used to assess cell viability, as per manufacturer’s protocol. Briefly, at each timepoint, cells were washed with HBSS and incubated with a solution containing 2 µM calcein-AM and 4 µM ethidium homodimer in HBSS for 30 min at 37 °C in a 5% CO_2_ humidified atmosphere. DMSO was added for positive dead cell controls. Excess dye was removed by washing with HBSS. Samples were visualised using an inverted IX73 Olympus microscope (Olympus Corporation, Tokyo, Japan). At least 5 regions of interest (ROI) per image were selected. Alive and dead cells were counted with ImageJ software (NIH, Bethesda, MD, USA) and percentage cell viability was calculated as alive cell number against total cell number.

### 2.9. Stem Cell Proliferation Analysis

Cell proliferation was assessed via nuclei counting. Briefly, cell cultures were washed with HBSS at each timepoint and fixed with a filtered (0.2 μm sterile syringe filter) solution of 2% paraformaldehyde (PFA) in PBS. Excess fixation solution was removed by PBS washes. Samples were then incubated for 5 min in a 4′,6-diamidino-2-phenylindole (DAPI, D1306, Invitrogen™, Gloucester, UK) in methanol at 0.5 μg/mL concentration solution. Excess DAPI solution was removed with PBS washes. Samples were visualised using an inverted IX73 Olympus microscope (Olympus Corporation, Tokyo, Japan). Cell nuclei were counted with ImageJ software (NIH, Bethesda, MD, USA); the watershed plugin was used to separate overlapped nuclei. At least 5 ROI per image were selected. Nuclei were counted to obtain cell number per area at the different timepoints.

### 2.10. Stem Cell Metabolic Activity Analysis

Cell metabolic activity was assessed using the alamarBlue™ (DAL1025, Invitrogen™, Gloucester, UK) assay, as per manufacturer’s protocol. Briefly, at the end of each culture timepoint, cells were washed with HBSS and 10% alamarBlue™ solution in HBSS was added. Samples were incubated for 3 h at 37 °C in a 5% CO_2_ humidified atmosphere. Absorbance at 550 nm excitation and 595 nm emission was then measured using a plate reader (ELx800, Biotek, Swindon, UK). Cell metabolic activity was expressed as the% reduction in the alamarBlue^®^ dye and normalised to the control group without MMC.

### 2.11. Gel Electrophoresis Analysis

At each timepoint collagen extraction from the cell layers and sodium dodecyl sulphate-polyacrylamide gel electrophoresis (SDS-PAGE) was performed [48]. Briefly, at each timepoint, cell culture media were removed and the cell layers were washed twice with HBSS at 37 °C. Cell layers were digested with pepsin from porcine gastric mucosa (P6887; 150 µL per well of a 24-well plate at a final concentration of 0.1 mg/mL in 0.05 M acetic acid) for 2 h at 37 °C under continuous shaking. Then, 100 µL of the digested cell layers were collected and neutralised with 20 µL of 0.1 N NaOH. Samples were either analysed immediately or stored at −80 °C until analysis. Digested and neutralised samples were appropriately diluted with distilled water and 5× sample buffer and heated at 95 °C for 5 min. Per well, 15 μL per sample solution was loaded on the gel (3% stacking gel/5% running gel). As a reference standard, 15 μL of 0.1 mg/mL in 0.05 M acetic acid bovine skin collagen type I (CBPE2US010, Symatese Biomateriaux, Chaponost, France) was used. Electrophoresis was performed in a Mini-PROTEAN Tetra Electrophoresis System (Bio-Rad, Watford, Hertfordshire, UK) by applying a potential difference of 50 mV for the initial 30–40 min and then 110 mV for the remaining time (50–70 min). The gels were washed with double-distilled water and stained using the PlusOne™ Silver Staining Kit, Protein (17115001, GE Healthcare, Chalfont St Giles, Buckinghamshire, UK), according to the manufacturer’s protocol. Images of the gels were taken after brief washing with double-distilled water. To quantify the cell-produced collagen type I deposition, the relative densities of collagen α1(I) and α2(I) chains were evaluated using ImageJ software (NIH, Bethesda, MD, USA) after applying the background subtraction plugin and correlated to the α1(I) and α2(I) chain bands densities of the reference standard collagen type I.

### 2.12. Immunocytochemistry Analysis

At each timepoint, media were removed, samples were washed with HBSS and fixed for 15 min with filtered 2% PFA at 4 °C. Excess PFA was removed with PBS washes. Blocking of non-specific bindings was performed with a 3% BSA (A7638) solution at room temperature for 30 min. The samples were then incubated for 90 min at room temperature with the following primary antibodies: mouse monoclonal anti-collagen type I (ab90395, 1:500 dilution), rabbit polyclonal anti-collagen type III (ab7778, 1:200 dilution) and rabbit polyclonal anti-collagen type IV (ab6586, 1:200 dilution); all Abcam, Ireland. The samples were washed 3 times with PBS and then incubated for 30 min with the following secondary antibodies at 1:500 dilution: goat anti-rabbit conjugated to AlexaFluor^®^ 488 (A-32731) and goat anti-mouse conjugated AlexaFluor^®^ 555 (A-32727); both ThermoFisher Scientific (Dublin, Ireland). As negative controls to subtract the background, the samples were incubated in PBS without primary antibodies and incubated with the corresponding secondary antibodies. Samples were washed 3 times with PBS and incubated for 5 min with DAPI at 0.5 μg/mL concentration in methanol. The samples were washed 3 times with PBS and mounted with 10 μL of VECTASHIELD^®^ HardSet™ Antifade Mounting Medium (H1500, Vector Laboratories, Burlingame, CA, USA). Glass coverslips with a diameter of 8 mm were placed on top of the samples. Images were taken with an Olympus IX73 inverted microscope (Olympus Corporation, Japan). The Q-Capture Pro 7 software (QImaging, Rockville, Maryland, USA) was used for image acquisition. At least 5 ROI per image were selected. Fluorescence intensity was calculated using the ImageJ software (NIH, Bethesda, MD, USA), subtracting from every ROI the average fluorescence intensity value of the negative control. The values were normalised to the corresponding cell number of each ROI.

### 2.13. Statistical Analysis

Numerical data are expressed as mean ± standard deviation (SD). Data were processed in Microsoft^®^ Excel and statistical analysis was performed using SigmaPlot^®^ 12.0 software (Systat Software, Slough, UK). One way analysis of variance (ANOVA) followed by pairwise multiple comparison (Holm-Sidak’s method) was used when the distributions of the populations were normal (Shapiro-Wilk normality test) and the variances of populations were equal (Levene’s test). Non-parametric analysis was performed using Kruskal-Wallis followed by Dunn’s multiple comparison post-hoc analysis for pairwise multiple comparisons, when the assumptions of parametric analysis were violated. Statistical significance was accepted at *p* < 0.05.

## 3. Results

### 3.1. Stem Cell Characterisation

The extracted cells were mesenchymal in origin as trilineage analysis (Appendix A) revealed calcium deposits in osteogenic media, lipid droplets in adipogenic media and glycosaminoglycan-rich ECM in chondrogenic media; FACS analysis (Appendix A) demonstrated that the cells were positive for CD90 (95.2% ± 0.3%), CD44 (58.4% ± 1.4%) and CD29 (41.7% ± 11.5%) and negative for CD105 (0.1% ± 0.0%) and CD34 (0.0% ± 0.0%); and immunocytochemistry analysis (Appendix A) visually corroborated the FACS analysis data.

### 3.2. HA Solubility Assessment

Solubility assessment in cell-free standard cell culture media at 37 °C (Appendix A) revealed that the 10 kDa HA was soluble up to 200 mg/mL; the 60 kDa HA was soluble up to 60 mg/mL; the 100 kDa HA was soluble up to 30 mg/mL; the 500 kDa HA was soluble up to 10 mg/mL; and the 1000 kDa HA was soluble up to 5 mg/mL. Subsequent bright field microscopy analysis after 4 days in culture (Appendix A) revealed that all concentrations of the 10 kDa HA, the 30 and 60 mg/mL concentrations of the 60 kDa HA and the 30 mg/mL concentration of the 100 kDa HA resulted in cell detachment, and the 20 mg/mL concentration of the 100 kDa HA resulted in microgel formation; thus, they were discarded for further analysis.

### 3.3. Dynamic Light Scattering Assessment

Zeta potential analysis (Table 1) revealed that all HA 60 kDa, HA 100 kDa (but the 10 mg/mL), HA 500 kDa (but the 10 mg/mL) and HA 1000 kDa concentrations exhibited a significantly (*p* < 0.05) lower negative charge than carrageenan; and that carrageenan and all HA 60 kDa, HA 100 kDa (but the 0.5 mg/mL), HA 500 kDa (but the 0.5 mg/mL) and HA 1000 kDa concentrations exhibited a significantly (*p* < 0.05) higher negative charge than Ficoll^®^. Within the different HA molecular weights, for HA 60 kDa and HA 1000 kDa, the charge was not affected (*p* > 0.05) as a function of increased concentration, whilst for HA 100 kDa and HA 500 kDa, the charge was significantly (*p* < 0.05) decreased as a function of increased concentration (Table 1).

Hydrodynamic radius analysis (Table 1) revealed that all concentrations of HA 60 kDa, the 5 and 10 mg/mL concentrations of HA 100 kDa and the 5 and 10 mg/mL concentrations of HA 500 kDa exhibited a significantly (*p* < 0.05) higher hydrodynamic radius than carrageenan; that the 0.5 mg/mL concentration of HA 100 kDa, the 0.5 and 1 mg/mL concentrations of HA 500 kDa and the 0.1 mg/mL concentration of HA 1000 kDa exhibited a significantly (*p* < 0.05) lower hydrodynamic radius than carrageenan; and that carrageenan and all concentrations of HA 60 kDa, HA 100 kDa, HA 500 kDa and HA 1000 kDa exhibited a significantly (*p* < 0.05) higher hydrodynamic radius than Ficoll^®^. Within the different HA molecular weights, for HA 60 kDa and HA 1000 kDa, the hydrodynamic radius was not affected (*p* > 0.05) as a function of increased concentration, whilst for HA 100 kDa and HA 500 kDa, the hydrodynamic radius was not significantly (*p* > 0.05) affected between 0.5 and 1 mg/mL concentration and was significantly (*p* < 0.05) increased as a function of increased (from 1 to 5 mg/mL and from 5 to 10 mg/mL) concentration (Table 1).

Polydispersity analysis (Table 1) revealed no significant differences in polydispersity between carrageenan, Ficoll^®^ and all concentrations of HA 60 kDa, HA 100 kDa (apart from the 10 mg/mL concentration which was the lowest (*p* < 0.05) of all), HA 500 kDa and HA 1000 kDa.

Fractional volume occupancy (%) analysis (Table 1) revealed that carrageenan exhibited a significantly (*p* < 0.05) higher % fractional volume occupancy than Ficoll^®^ and all HA 60 kDa concentrations, the 1, 5 and 10 mg/mL HA 100 kDa concentrations, the 5 and 10 mg/mL HA 500 kDa concentrations and the 5 mg/mL HA 1000 kDa concentration exhibited a significantly (*p* < 0.05) higher % fractional volume occupancy than carrageenan and Ficoll^®^. Within the different HA molecular weights, the highest concentration of HA 60 kDa (10 mg/mL), HA 100 kDa (10 mg/mL), HA 500 kDa (10 mg/mL) and HA 1000 kDa (5 mg/mL) exhibited the highest (*p* < 0.05) % fractional volume occupancy (Table 1). Among all groups, the 10 mg/mL HA 100 kDa and the 0.1 mg/mL HA 1000 kDa exhibited the highest (*p* < 0.05) and the lowest (*p* < 0.05) % fractional volume occupancy, respectively (Table 1).

### 3.4. Cell Morphology Assessment

Bright field microscopy analysis (Appendix A) revealed no apparent differences at any timepoint in cell morphology as a function of MMC, independently of the MMC agent used and its respective concentration.

### 3.5. Cell Viability, Proliferation and Metabolic Activity Assessment

Qualitative (Appendix A) and quantitative (Appendix A) cell viability analysis revealed no apparent differences at any timepoint in cell viability as a function of MMC, independently of the MMC agent used and its respective concentration.

Quantitative cell proliferation (Appendix A) and metabolic activity (Appendix A) analyses revealed that the Ficoll^®^ at all timepoints and the carrageenan, the 10 mg/mL HA 100 kDa, the 5 and 10 mg/mL HA 500 kDa and the 5 mg/mL HA 1000 kDa at day 8 significantly (*p* < 0.05) decreased cell number and increased cell metabolic activity.

### 3.6. Gel Electrophoresis Assessment

SDS-PAGE and complementary densitometry analyses (Figure 1) revealed that carrageenan induced the highest (*p* < 0.05) collagen type I deposition at all timepoints and Ficoll^®^ induced significantly (*p* < 0.05) higher collagen type I deposition than the non-MMC group only at day 4 and in some instances, at day 8, induced the lowest (*p* < 0.05) collagen type I deposition. The high concentrations (5 and 10 mg/mL) of HA 60 kDa (Figure 1) and HA 100 kDa (Figure 1) induced significantly (*p* < 0.05) higher collagen type I deposition than the non-MMC group only at day 4. The 5 mg/mL concentration of HA 500 kDa (Figure 1) induced significantly (*p* < 0.05) higher collagen type I deposition than the non-MMC group only at day 4. None of the HA 1000 kDa (Figure 1) concentrations induced significantly (*p* > 0.05) higher collagen type I deposition than the non-MMC group at any timepoint.

### 3.7. Immunocytochemistry Assessment

Immunocytochemistry (Figure 2) and complementary fluorescence intensity (Appendix A) analyses for collagen type I revealed that carrageenan induced the highest (*p* < 0.001) collagen type I deposition at all timepoints, Ficoll^®^ induced higher (*p* < 0.05) collagen type I deposition than the non-MMC group only at days 4 and 6 and HA had no effect (*p* > 0.05) in collagen type I deposition at any timepoint, independently of the concentration and molecular weight.

Immunocytochemistry (Figure 3) and complementary fluorescence intensity (Appendix A) analyses for collagen type III revealed that Ficoll^®^ induced the highest (*p* < 0.001) collagen type III deposition (at day 6, the 10 mg/mL HA 500 kDa group was not significantly (*p* > 0.05) different to the Ficoll^®^ group) and the carrageenan, the 10 mg/mL HA 60 kDa, the 5 mg/mL HA 100 kDa (only at day 6), the 10 mg/mL HA 100 kDa, the 5 and 10 mg/mL HA 500 kDa and the 5 mg/mL HA 1000 kDa groups induced higher (*p* < 0.05) collagen type III deposition than the non-MMC group at all timepoints.

Immunocytochemistry (Figure 4) and complementary fluorescence intensity (Appendix A) analyses for collagen type IV revealed that Ficoll^®^ induced the highest (*p* < 0.001) collagen type IV deposition and the carrageenan, the 10 mg/mL HA 60 kDa, the 10 mg/mL HA 100 kDa, the 10 mg/mL HA 500 kDa (only at days 6 and 8) and the 5 mg/mL HA 1000 kDa (only at days 6 and 8) groups induced higher (*p* < 0.05) collagen type IV deposition than the non-MMC group at all timepoints.

## 4. Discussion

In recent years, the use of MMC has been advocated for, for the accelerated development of tissue moduli for regenerative medicine [49] and drug discovery [50] purposes, as well as for the development of cell-derived matrices for effective cell expansion ex vivo [51] and as a quality control tool in cell culture media development [52]. Despite these significant strides, the optimal (with respect to the highest ECM deposition in the shortest period of time) MMC agent is still elusive. Among the various MMC agents, carrageenan has been comprehensively shown to induce the highest ECM deposition in the shortest period of time due to its negative charge and high polydispersity [13,14,15]. However, at the time of writing, carrageenan is only FDA-approved as a food additive [53]. In order, therefore, to capitalise on the immense power of MMC to accelerate the development of implantable tissue engineering devices, regulatory-compliant for medical applications macromolecules should be identified. With these in mind, herein we ventured to assess the potential of different concentrations of different molecular weight HAs as MMC agents. It is worth noting that the safety and efficacy of HA has been well-demonstrated in clinical practice (e.g., intra-articular viscosupplementation [54], aesthetic reconstruction of interdental papilla loss in anterior teeth [55], intra-dermal injections [56] and as dermal filler in photodamaged skin [57]) and it has been regulatory cleared (e.g., topical wound cream: 510(K) number: K172747 and treatment of pain in osteoarthritis: premarket approval number: P170016).

Starting with solubility assessment, we found a molecular weight-dependent solubility, with the 1000 kDa molecular weight HA able to remain soluble up to 5 mg/mL concentration, whilst the 10 kDa molecular weight HA was found to be soluble up to 200 mg/mL concentration. In accordance with our work, similar HA concentrations/molecular weights have been used in the literature as media supplements (e.g., 2 mg/mL 1800 kDa HA in human bone marrow stem cells [58]; 0.25–1.00 mg/mL 600–2000 kDa HA in human tenocyte cultures [59,60]; 0.5 and 5 mg/mL 1500 kDa HA in human neonatal fibroblast cultures [36]; 0.5, 1.0 and 2.0 mg/mL 30–1,300 kDa HA in calvarial mesenchymal cell (from 13 days old mouse embryos) cultures [61]; 0.1 and 0.5 mg/mL 500–730 kDa HA in human chondrocytes [62]; 0.5, 1 and 2 mg/mL 60 kDa and 900 kDa HA in rat calvarial-derived cell cultures [63]; 0.1 mg/mL 2000 and 3000 kDa HA in equine bone marrow stem cell chondrogenic differentiation cultures [64]; and 4 mg/mL 800 kDa HA in porcine bone marrow stem cell cultures [65]).

With respect to biophysical analysis, all molecules assessed had a negative charge; the high negative charge of carrageenan can be attributed to its sulphate content [66], and the Ficoll^®^, although it is considered to be non-ionic, is a rather heterogeneous in molecular weight macromolecule and has been demonstrated to contain negatively charged residues on about 50% of its macromolecular components [67], and the carboxyl groups of the glucuronic acid residues are negatively charged at physiological pH and ionic strength, making hyaluronic acid a negatively charged polyanionic macromolecule [68,69,70]. For the HA 100 kDa and the HA 500 kDa, a concentration-dependent zeta potential decrease and a hydrodynamic radius increase was observed as a function of increased concentration. This is in agreement with previous publications, where the charge was decreased and the mean particle size was increased as a function of increasing nanoparticle concentrations due to their aggregation in high concentrations [71,72,73]. Carrageenan, Ficoll^®^, HA 60 kDa, HA 100 kDa (only the 1 mg/mL concentration), HA 500 kDa and HA 1000 kDa exhibited a polydispersity index of > 0.7, which indicates a high degree of heterogenicity, due to broad size (i.e., polydisperse) distribution or agglomeration or aggregation of the particles of a population [74]. We attribute the observed differences in the polydispersity index of HA 100 kDa to experimental limitations of the method, as high sample concentrations (and therefore too high particle concentrations) result in multi-scattering or unpredictable agglomeration, whilst low sample concentrations (and therefore too dilute particle concentrations) may not generate enough light to analyse [72,73]. With respect to % fractional volume occupancy, only the 0.5 and 1 mg/mL HA 500 concentrations and the 0.1 mg/mL HA 1000 concentration exhibited a % fraction volume occupancy below 60%, which is at the boundary of probability. Over the years, a diverse range of % fraction volume occupancy values have been reported in the literature using the method that we also used herein [47] (e.g., 5.2% for dextran sulphate 500 kDa [75]; 28% for a cocktail of dextran sulphate 10 kDa, Ficoll^®^ 70 kDa and Ficoll^®^ 400 kDa [51]; 9–54% for a Ficoll^®^ 70 kDa and Ficoll^®^ 400 kDa cocktail (subject to the concentration of each molecule in the solution) and polyvinylpyrrolidone 40 kDa and polyvinylpyrrolidone 360 kDa (subject to the concentration of each molecule in the respective solution) [32]; and >100% for highly sulphated seaweed polysaccharides, such as carrageenan, fucoidan, galactofucan, arabinogalactan and ulvan [15]). Although % fractional volume occupancy is frequently employed to predict the effectiveness of a macromolecule to occupy space/exclude volume in cell culture and consequently result in enhanced and accelerated ECM deposition, we believe that this approach should be treated with caution as the % fractional volume occupancy is calculated from the molecular weight and the hydrodynamic radius, based on the assumption that the molecule in question is spherical, which is not the case in real life (e.g., dextran is a ribbon-like or rod-like molecule, and Ficoll^®^ is considered as a deformable sphere, as opposed to a compact sphere [76]).

Moving into basic cellular function analysis, none of the MMC agents assessed affected cell morphology and viability, whilst the Ficoll^®^ at all timepoints, the carrageenan at day 8, the 10 mg/mL HA 100 kDa at day 8, the 10 mg/mL HA 500 kDa at day 6 and 8 and the 5 mg/mL HA 1000 kDa at day 8 induced reduced cell proliferation and increased metabolic activity in comparison to the non-MMC control group. Previous studies have shown Ficoll^®^ and/or carrageenan to not affect human bone marrow stem cell morphology, viability and metabolic activity [49,77] (albeit, a non-significant decrease in metabolic activity has been reported for carrageenan [78] and a significant increase in proliferation has been reported for Ficoll^®^ after 24 h [79], which may be asymptomatic), whilst in human adipose-derived stem cell (ADSC) cultures, carrageenan has been shown to not affect cell morphology and viability, but to increase cell proliferation and to decrease cell metabolic activity [15], and Ficoll^®^ has been shown to affect cell morphology, to significantly increase metabolic activity only at day 1 and to significantly decrease DNA at days 4, 7 and 11 [80], in all cases, in comparison to the respective non-MMC group. In permanently differentiated cell cultures, Ficoll^®^ has been shown to not affect the morphology, viability and metabolic activity of human corneal fibroblasts [31] and carrageenan has been shown to not affect the morphology, viability, metabolic activity and proliferation of human corneal fibroblasts [81] and human tenocytes [82]. With respect to HA, 2 mg/mL 1800 kDa HA supplementation in human bone marrow stem cells has been shown to not affect DNA content [58] and 1 μg/mL 850 kDa HA supplementation in human ADSCs has been shown to significantly increase cell proliferation (the 0.1 and 0.3 μg/mL concentrations; although they increased cell proliferation, the increase was not significant) and the 5 mg/mL concentration has been shown to decrease metabolic activity only at day 7 (longest timepoint assessed) and to not affect cell viability at any timepoint [83]. In permanently differentiated cell cultures, 0.25–1.00 mg/mL 600–2000 kDa HA supplementation has been shown to not affect human tenocyte morphology and to increase viability and proliferation in a dose-dependent manner [59,60]; 0.01, 0.1 and 1 mg/mL 800,000 viscometric average molecular weight (11.8–19.5 dL/g viscosity) HA supplementation has been shown to not affect rabbit chondrocyte (embedded in a collagen hydrogel) morphology and to increase cell proliferation (0.1 mg/mL HA induced the highest proliferation) [84]; 0.5 and 5 mg/mL 1500 kDa HA supplementation has been shown to not affect human neonatal fibroblast proliferation [36]; and 0.5–2 mg/mL 60 kDa HA and 0.5–1 mg/mL 900 kDa HA increased proliferation of calvarial rat-derived cells [63]). It is also worth noting that HA supplementation has also been shown to inhibit cell proliferation (e.g., 1–100 µg/mL 1000 and 1800 kDa HA negatively affected the proliferation of foetal rabbit skin fibroblasts [85] and rabbit synovial cells [86]; and 2 mg/mL 500, 800, 1600 and 3600 kDa inhibited rabbit tendon cell proliferation [87]). Collectively, although a cell-dependent response to MMC is evidenced, an adverse effect in basic cellular functions cannot be noted.

SDS-PAGE and immunocytochemistry analyses revealed that the eADSCs were able to synthesise and deposit collagen types I, III and IV. All these ECM macromolecules are essential for physiological tissue development, function and healing; for example, collagen types I and III are the most abundant collagens in the human body, with collagen type I playing a crucial role in the structural and biomechanical integrity of tissues [88] and collagen type III regulating collagen type I fibrillogenesis and fibril diameter [89]. Collagen type IV is the most copious collagen in basement membranes and plays a crucial role in embryogenesis and wound healing, whilst minor structural differences can lead to pathophysiologies [90]. Under MMC conditions, carrageenan induced the highest collagen type I deposition, which is in agreement with previous publications with a diverse range of cell populations (e.g., in human skin fibroblasts over Ficoll^®^ 70 kDa, Ficoll^®^ 400 kDa, Ficoll^®^ 1000 kDa and cocktails thereof, dextran sulphate 10 kDa, dextran sulphate 100 kDa, dextran sulphate 500 kDa and cocktails thereof [14]; in human corneal fibroblasts over dextran sulphate 500 kDa [81]; in human lung and skin fibroblasts over dextran sulphate 500 kDa and a Ficoll^®^ 70 kDa and Ficoll^®^ 400 kDa cocktail [13]; in human bone marrow stem cells over a Ficoll^®^ 70 kDa and Ficoll^®^ 400 kDa cocktail [49]; and in human ADSCs over fucoidan, galactofucan, arabinogalactan and ulvan [15]). This superiority of carrageenan in inducing the highest ECM deposition has been attributed to its negative charge and high polydispersity [13,14]. With respect to Ficoll^®^, the highest (over the non-MMC control) collagen type I deposition was only observed at day 4. Although one study has shown the Ficoll^®^ cocktail to result in lower collagen deposition than the non-MMC group after 14 days in human neonatal fibroblast cultures [36], most studies have shown the Ficoll^®^ cocktail to enhance collagen type I deposition in various cell populations (e.g., human skin fibroblasts [91], human corneal fibroblasts [31], human lung and skin fibroblasts [13], human bone marrow stem cells [79] and human ADSCs [80]), albeit at slower rates than sulphated polysaccharides [14,50]. Finally, HA had no effect on collagen type I deposition at any timepoint, independently of the concentration and molecular weight. With respect to the influence of increasing concentrations of a crowder in collagen deposition, previous studies have shown collagen deposition to be increased as a function of increasing carrageenan (for example) concentration (especially at early time points) [13,14,49]; above the optimal (with respect to enhanced and accelerated collagen deposition) concentration range, the concomitant increase in viscosity reduces reaction kinetics, as observed in protein folding/diffusion studies [92,93]. Thus, one would have expected increasing HA concentration, which resulted in an increased % fractional volume occupancy, to also result in increased collagen deposition. Considering though that polydispersity was not increased as a function of increasing HA concentration, and neither was the collagen deposition, we feel that this further substantiates our aforementioned claim with respect to the theoretical, as opposed to practical, nature of % fractional volume occupancy. With respect to the ability of HA to enhance and accelerate collagen type I deposition, our data are in both agreement (e.g., 10 mg/mL HA 2, 250 and 1000 kDa injected into full-thickness dermal wounds of aged mice did not increase collagen type I deposition [94], 50 mg/mL HA-12 saccharide units derived from HA 1700 kDa did not affect collagen synthesis in human skin fibroblasts [95], 100 mg/mL had no effect in collagen synthesis in foetal rabbit skin fibroblasts [85], 500 mg/mL HA (molecular weight was not stated) had no effect on collagen synthesis in human skin fibroblast cultures [96], 500 mg/mL HA 1600 kDa had no effect on the cell layer or the media collagen content in human skin fibroblast cultures [97], 500 and 5000 mg/mL HA 1500 kDa had a marginal (assessed via electrophoresis and Raman imaging) and no effect (assessed via immunocytochemistry), respectively, in collagen deposition in human neonatal fibroblast cultures [36]) and disagreement (e.g., 1 and 10 mg/mL HA 1000–1800 kDa increased collagen synthesis in foetal rabbit skin fibroblasts [85], and 1 and 10 mg/mL HA-12 saccharide units derived from HA 1700 kDa increased collagen synthesis in human skin fibroblasts [95]) with previous observations. With respect to collagen type III and collagen type IV deposition, the Ficoll^®^, carrageenan, 10 mg/mL HA 60 kDa, 10 mg/mL HA 100 kDa, 10 mg/mL HA 500 kDa, and 5 mg/mL HA 1000 kDa induced their highest deposition. Enhanced collagen type III and/or collagen type IV deposition with Ficoll^®^ and/or carrageenan has been reported previously in human bone marrow stem cells [49,78], human corneal fibroblasts [31,81], human skin fibroblasts [13,98] and human tenocyte [82,99] cultures, but not in human ADSCs [15], human tenocytes, neonatal fibroblasts, adult fibroblasts, bone marrow stem cells [77] and human neonatal fibroblast [36] cultures. With respect to HA, 1, 10 and 50 mg/mL HA 1700 kDa and HA-12 and HA-880 saccharide units derived from HA 1700 kDa increased collagen type III synthesis in human skin fibroblast cultures [95], whilst 500 and 5000 mg/mL HA 1500 kDa had no effect on collagen type III and collagen type IV deposition in human neonatal fibroblast cultures [36]. It is also worth noting that HAs of 2300 and 3000 to 5800 kDa had an inhibitory/cytotoxic effect on the monolayer in human skin fibroblast cultures, whilst 1000 mg/mL HA 500 to 1200 kDa increased collagen type III synthesis in human skin fibroblasts embedded within a collagen hydrogel, as was evidenced with Sirius red staining [100]. Further, 10 mg/mL HA 250 kDa increased collagen type III deposition in full-thickness dermal wounds of aged mice, whilst 10 mg/mL HA 2 and 1000 kDa had no effect [94]. It appears that a concentration cut-off exists in the literature for the high molecular weight HAs, with concentrations up to 50 mg/mL able to enhance collagen deposition, whilst concentrations higher than 50 mg/mL have no effect. In our case, the lowest concentration assessed was 100 mg/mL for all HAs molecular weights (60, 100, 500 and 1000 kDa). One can argue that concentrations below 50 mg/mL are outside of the MMC zone, but considering the very high water-binding capacity of HA (up to 6 L per g [37,38]), it may be actually possible to act as an MMC agent even at such low concentrations. One cannot also exclude that cell-specific biological events may be at play, considering that, through growth factor interaction and involvement in signalling cascades, sulphated polysaccharides have been shown to enhance the chondrogenic and osteogenic potential of stem cells [15,49,78], whilst non-sulphated polysaccharides have been shown to enhance the adipogenic potential of stem cells [101,102]. Biochemical interactions can also be responsible for the observed results, as previous studies have shown mutual tropocollagen/HA steric exclusion [103,104].

## 5. Conclusions

In the quest for the optimal macromolecular crowder for enhanced and accelerated extracellular matrix deposition, herein we ventured to assess the potential of different concentrations and molecular weight hyaluronic acids. As carrageenan and Ficoll^®^, customarily used crowding molecules, induced the highest collagen I, collagen III and collagen IV deposition, unless a biological benefit for hyaluronic acid is identified, we recommend their use as macromolecular crowding molecules for enhanced and accelerated extracellular matrix deposition.

## Figures and Tables

**Figure 1 cells-10-00859-f001:**
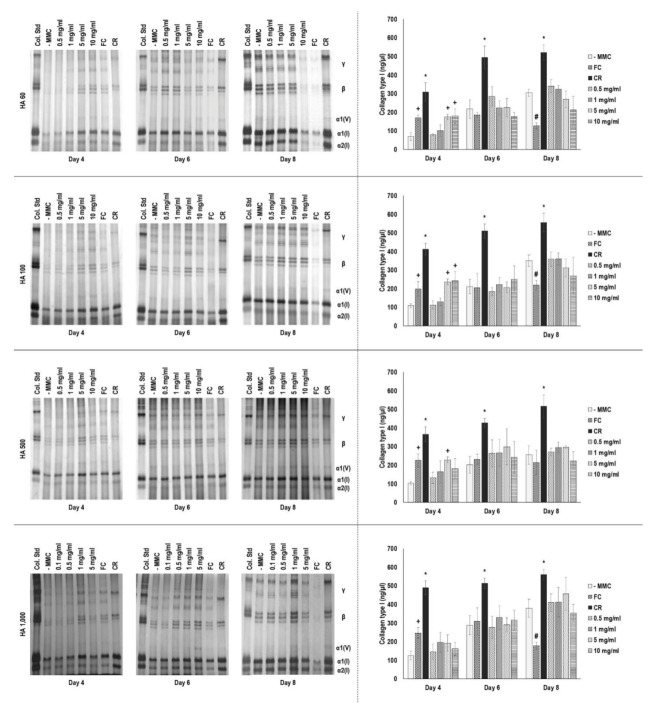
SDS-PAGE and complementary densitometry analyses revealed that carrageenan induced the highest (*p* < 0.05) collagen type I deposition at all timepoints. * indicates significantly (*p* < 0.05) higher deposition among all groups at a given timepoint. + indicates significantly (*p* < 0.05) higher deposition than the non-macromolecular crowding (MMC) control group at a given timepoint. # indicates significantly (*p* < 0.05) lower deposition than the non-MMC control group at a given timepoint.

**Figure 2 cells-10-00859-f002:**
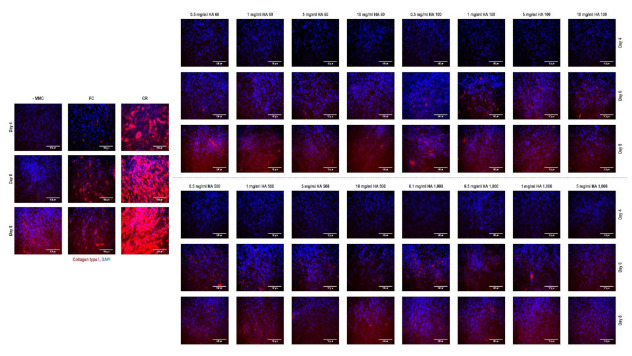
Immunocytochemistry analysis revealed that carrageenan induced the highest collagen type I deposition at all timepoints and at day 8 (longest timepoint assessed), no differences were observed between the non-MMC group and the Ficoll^®^ and any of the HA groups.

**Figure 3 cells-10-00859-f003:**
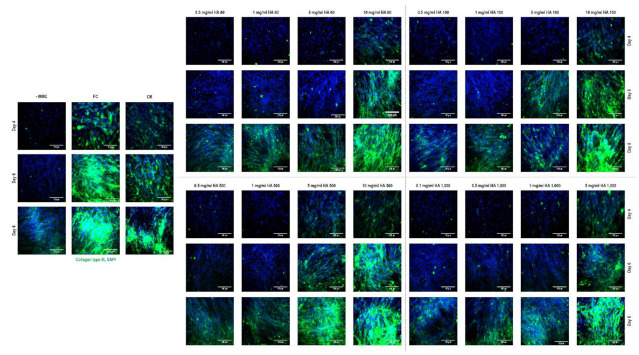
Immunocytochemistry analysis revealed that Ficoll^®^ induced the highest collagen type III deposition at all timepoints and at day 8 (longest timepoint assessed), the carrageenan, the 10 mg/mL HA 60 kDa, the 10 mg/mL HA 100 kDa, the 5 and 10 mg/mL HA 500 kDa and the 5 mg/mL HA 1000 kDa induced significantly higher collagen type III deposition than the non-MMC group.

**Figure 4 cells-10-00859-f004:**
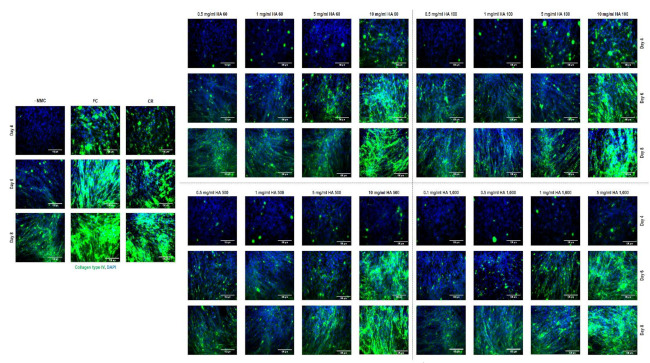
Immunocytochemistry analysis revealed that Ficoll^®^ induced the highest collagen type IV deposition at all timepoints and at day 8 (longest timepoint assessed), the carrageenan, the 10 mg/mL HA 60 kDa, the 10 mg/mL HA 100 kDa, the 10 mg/mL HA 500 kDa and the 5 mg/mL HA 1000 kDa induced significantly higher collagen type IV deposition than the non-MMC group.

**Table 1 cells-10-00859-t001:** Dynamic light scattering analysis.

Polymer (Concentration)	Zeta Potential (mV)	Hydrodynamic Radius (nm)	Polydispersity Index	Fractional Volume Occupancy (%)
FC 70 (25 mg/mL) + FC 400 (37.5 mg/mL)	−2.2 ± 1.1 #	15 ± 1 #	0.77 ± 0.11	(111 ± 13)
CR (75 µg/mL)	−26.5 ± 6.7 *	140 ± 29	0.68 ± 0.11	(1019 ± 570)
HA 60 (0.5 mg/mL)	−11.8 ± 1.9	246 ± 68	0.88 ± 0.12	(36,140 ± 29,701)
HA 60 (1 mg/mL)	−10.8 ± 1.6	302 ± 60	0.89 ± 0.12	(124,894 ± 61,980)
HA 60 (5 mg/mL)	−13.3 ± 0.8	249 ± 133	0.87 ± 0.11	(516,151 ± 631,032)
HA 60 (10 mg/mL)	−13.1 ± 2.7	262 ± 108	0.86 ± 0.12	(995,364 ± 771,048 +)
HA 100 (0.5 mg/mL)	−2.6 ± 2.3 #	85 ± 10	0.51 ± 0.06	(784 ± 264)
HA 100 (1 mg/mL)	−12.8 ± 1.6	194 ± 113	0.68 ± 0.37	(5448 ± 116)
HA 100 (5 mg/mL)	−16.0 ± 4.9	570 ± 17	0.60 ± 0.11	(2,341,046 ± 211,149)
HA 100 (10 mg/mL)	−23.2 ± 2.1 *, +	1886 ± 123 *, +	0.35 ± 0.02 #	(170,651,055 ± 33,934,948 *, +)
HA 500 (0.5 mg/mL)	−3.7 ± 0.7 #	59 ± 15	0.86 ± 0.12	59 ± 37
HA 500 (1 mg/mL)	−8.8 ± 2.6	46 ± 8	0.92 ± 0.08	51 ± 26
HA 500 (5 mg/mL)	−18.0 ± 3.1	213 ± 36	1.00 ± 0.00	(25,933 ± 12,345)
HA 500 (10 mg/mL)	−25.8 ± 3.3 *, +	604 ± 68 +	1.00 ± 0.00	(1,140,733 ± 384,037 +)
HA 1000 (0.1 mg/mL)	−7.1 ± 4.0	70 ± 24	0.97 ± 0.03	11 ± 8 #
HA 1000 (0.5 mg/mL)	−6.9 ± 4.2	99 ± 53	0.96 ± 0.08	(193 ± 226)
HA 1000 (1 mg/mL)	−9.0 ± 2.5	95 ± 29	0.78 ± 0.18	(256 ± 204)
HA 1000 (5 mg/mL)	−12.4 ± 1.5	133 ± 40	0.64 ± 0.09	(3484 ± 2294 +)

All molecules were negatively charged and, among all molecules and concentrations, the FC and the 0.5 mg/mL HA 60 kDa and HA 100 kDa exhibited the highest (*p* < 0.05) charge. The 10 mg/mL HA 100 kDa and the Ficoll^®^ exhibited the highest (*p* < 0.05) and the lowest (*p* < 0.05) hydrodynamic radius, respectively. The 10 mg/mL HA 100 kDa exhibited the lowest (*p* < 0.05) polydispersity index. The 10 mg/mL HA 100 kDa and the 0.1 mg/mL HA 1000 kDa exhibited the highest (*p* < 0.05) and the lowest (*p* < 0.05) % fraction volume occupancy, respectively. * indicates significantly (*p* < 0.05) higher values among all groups. + indicates significantly (*p* < 0.05) higher values within a given HA molecular weight. # indicates significantly (*p* < 0.05) lower values among all groups. In parentheses, the calculated % fractional volume occupancy values, but as the values are > 100%, they cannot be considered as pragmatic. FC: Ficoll^®^. CR: Carrageenan. HA: Hyaluronic acid

## Data Availability

Data are available from DIZ upon request.

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
