# Peer review of "Hyaluronic Acid as Macromolecular Crowder in Equine Adipose-Derived Stem Cell Cultures"

_cells, 2021, doi:10.3390/cells10040859_

Round 1

Reviewer 1 Report

The authors present a study in which they characterise hyaluronic acid (HA) as a potential use in macromolecular crowding experiments. All experimental results are compared to the widely used crowing agents Ficoll and carrageenan.

The subject of macromolecular crowding always involves the search of an ideal crowder. Such extensive, fundamental investigations are scientifically crucial to differentiate between excluded volume effects and specific interactions.

In the present manuscript hyaluronic acid is characterised for its size, charge and volume first. The major work dwells on the effect of the crowder on stem cell cultures.

I have major critics with respect to the characterisation of HA. The method (DLS) or the machinery used seem not to work well enough to provide reliable data. The authors should repeat the measurements or use other techniques.

However, the extensive work on the effects of HA on cell lines are reliable and make this work worth to be published. Unfortunately, hyaluronic acid cannot be surpass the existing crowders, but who knows if is might be of use for different systems in future.

Author Response

Reviewer 1 Comments

Comment 1: I have major critics with respect to the characterisation of HA. The method (DLS) or the machinery used seem not to work well enough to provide reliable data. The authors should repeat the measurements or use other techniques.

Response: We would like to thank the reviewer for the overall positive feedback on our work. We appreciate the reviewer’s concern with respect to the accuracy of the DLS method. We agree that nanoparticle tracking analysis is far more accurate, as we have shown before (Satyam et al. Macromolecular crowding meets tissue engineering by self-assembly: A paradigm shift in regenerative medicine. Adv Mater. 2014;26(19):3024-34). Unfortunately, we do not have access to this piece of equipment in our laboratory and we were / are not able to outsource it at the moment due to the Covid19 situation. Nonetheless, we would like to mention that DLS is used extensively in the field to assess the biophysical properties of the crowding molecules. We have also made sure that the limitations of the method are mentioned in the manuscript. For example, we mention the following in the discussion section, where the biophysical properties are discussed:

We attribute the observed differences in polydispersity index of HA 100 kDa to experimental limitations of the method, as high sample concentrations (and therefore too high particle concentrations) result in multi-scattering or unpredictable agglomeration, whilst low sample concentrations (and therefore too dilute particle concentrations) may not generate enough light to analyse [72,73].

Although % fractional volume occupancy is frequently employed to predict the effectiveness of a macromolecule to occupy space / exclude volume in cell culture and consequently result in enhanced and accelerated ECM deposition, we believe that this approach should be treated with caution as the % fractional volume occupancy is calculated from the molecular weight and the hydrodynamic radius, based on the assumption that the molecule in question is spherical, which is not the case in real life (e.g. dextran is ribbon-like or rod-like molecule, Ficoll® is considered as a deformable sphere, as opposed to a compact sphere [76]).

We are reluctant to completely remove the biophysical assessment from the manuscript. We believe that it adds value, despite the limitations. If, however, the reviewer feels strongly about this, we can put it in the supplementary information?

Reviewer 2 Report

The authors analysed the influence of different concentrations of different molecular weights hyaluronic acid (HA) on equine adipose-derived stem cell cultures. They showed that 10 mg/ml 100 kDa and 500 kDa HAs exhibited the highest negative charge and hydrodynamic radius and the 10 mg/ml 100 kDa HA exhibited the lowest polydispersity index and the highest % fraction volume occupancy.  Results are clear and well described. Still, some issues must be considered before publication.

The abstract is somehow misleading -  define if HA have any advantages over customarily used crowding molecules
Divide stem cells characterization into separate sections - it is illegible in such a form
Fig  S1 – how was viscosity analysed?
Fig S2 and S3 - standardize photo formatting – grayscale?
Fig S4 and SB - the description lacks any development of the abbreviations used
Discussion is too long and wordy – consider rewriting focusing on the most important aspects 

Author Response

Reviewer 2

Comment 1: The abstract is somehow misleading - define if HA have any advantages over customarily used crowding molecules.

Response: We have edited the abstract, which reads as follows in the revised version of the manuscript: ‘Although HA had the potential to act as macromolecular crowding agent, it did not outperform carrageenan and Ficoll®, the most widely used macromolecular crowding molecules, in enhanced and accelerated collagen I, collagen III and collagen IV deposition.’ We have also edited the conclusions section, which reads as follows in the revised version of the manuscript: ‘As carrageenan and Ficoll®, customarily used crowding molecules, induced the highest collagen I, collagen III and collagen IV deposition, unless a biological benefit for hyaluronic acid is identified, we recommend their use as macromolecular crowding molecules for enhanced and accelerated extracellular matrix deposition.

Comment 2: Divide stem cells characterization into separate sections - it is illegible in such a form.

Response: We have used the following introductory sentence at section 2.4. Stem cell characterisation analysis in the revised version of the manuscript: ‘Multipotency was assessed with established osteogenic, adipogenic and chondro-genic protocols, using cells in DMEM as negative controls, and surface markers were assessed with FACS analysis, validating the antibodies with immunocytochemistry.’ We then used the following 4 subheadings in the revised version of the manuscript: ‘2.4.1. Osteogenic analysis’, ‘2.4.2. Adipogenic analysis’, ‘2.4.3. Chondrogenic analysis’ and ‘2.4.4. FACS analysis

Comment 3: Fig S1 – how was viscosity analysed?

Response: We did not assess viscosity. We are mentioning in Table S1 ‘Too viscus / Gel’, but in most cases it was a microgel. We have edited the legend of Table S1, which reads as follows in the revised version of the manuscript: ‘Table S1: Qualitative HA solubility assessment as a function of molecular weight and concertation. ✓ indicates soluble

Comment 4: Fig S2 and S3 - standardize photo formatting – grayscale?

Response: We believe that the reviewer is referring to Figure S3 and S4, as in Figure S2, we used dyes (alizarin red S stain, red oil O stain and Alcian blue) and we feel that grayscale will not be accurate. We have revised (grayscale) Figure S3 and Figure S4, as per reviewer’s suggestion, for consistency.

Comment 5: Fig S4 and SB - the description lacks any development of the abbreviations used

Response: The full legends of the supplementary figures somehow did not come into the manuscript. Maybe the online system has a word limit? The full legends have now been added in the revised version of the manuscript and are provided herein as well: ‘Table S1: Qualitative HA solubility assessment as a function of molecular weight and concertation. ✓ indicates soluble; Figure S1: Experimental workflow of cell culture work and analysis; Figure S2: The extracted cells had the capacity to deposit calcium in osteogenic media, lipid droplets in adipogenic media and glycosaminoglycans in chondrogenic media (A). FACS analysis revealed that 95.2 % ± 0.3 % of the cells were positive for CD90, 58.4 % ± 1.4 % of the cells were positive for CD44 and 41.7 % ± 11.5 % of the cells were positive for CD29 and 0.1 % ± 0.0 % and 0.0 % ± 0.0 % of the cells were negative for CD105 and CD34, respectively (B). Immunocytochemistry analysis visually verified the FACS analysis data (C); Figure S3: Bright field microscopy analysis after 4 days in culture revealed that up to 10 mg/ml concentrations of the HAs 60 kDa, 100 kDa and 500 kDa and up to 5 mg/ml concentrations of HA 1,000 kDa were completely soluble and allowed cell attachment and growth (black font). All concentration of HA 10 kDa and concentrations higher than 10 mg/ml of HA 60 kDa and 100 kDa (red font) resulted in cell detachment or microgel formation; Figure S4: Bright field microscopy analysis revealed no apparent differences at any timepoint in cell morphology between the groups; Figure S5: Qualitative cell viability analysis revealed no apparent differences at any timepoint in cell viability between the groups; Figure S6: Quantitative cell viability analysis revealed no apparent differences at any timepoint in cell viability between the groups (A). Quantitative cell proliferation (B) and metabolic activity (C) analyses revealed that the Ficoll® at all timepoints significantly (p < 0.05) decreased cell number and increased cell metabolic activity. * indicates significant (p < 0.05) difference to the - MMC control group at a given timepoint; Figure S7: Fluorescence intensity analyses revealed that carrageenan induced the highest (p < 0.001) collagen type I deposition (A), Ficoll® induced the highest (p < 0.001) collagen type III deposition [at day 6, the 10 mg/ml HA 500 kDa group was not significantly (p > 0.05) different to the Ficoll® group] (B) and Ficoll® induced the highest (p < 0.001) collagen type IV deposition (C) than the non MMC group at all timepoints. * indicates significantly (p < 0.05) higher to the - MMC control group at a given timepoint. + indicates significantly (p < 0.05) higher to all groups at a given timepoint.

Comment 6: Discussion is too long and wordy – consider rewriting focusing on the most important aspects

Response: We appreciate that the discussion is rather detailed. However, we feel that as this is the first comprehensive study of the use of hyaluronic acid as macromolecular crowding agent, all sections are important. If the reviewer though feels strongly about this, we can transfer some elements (e.g. cell viability, metabolic activity, proliferation and morphology) into the supplementary information.

Round 2

Reviewer 1 Report

A molecular characterization of hyaluronic acid (HA) is crucial for a sound scientific work that attempts to address molecular crowding effects and not some kind of specific interaction. Such a characterization allows to relate physical properties, e.g. excluded volume and size to a cellular response. As long as these physical parameters are not known, all of the study is not scientifically meaningful enough for publication. It lacks basic understanding of the observed effects. Is the HA interaction dominated by excluded volume effects (macromolecular crowding) or specific interactions? Both is possible.

The authors conclude already that macromolecular crowding is the driving force (title), but they cannot convince with the work as it is presented.

Hence, I feel obliged to point out again, that the characterization of HA (Fig. 1) is severely WRONG. This issue should not be “hidden” in the supplemental nor addressed in a rather handwaving discussion only.

Just to give examples:

For HA60 (60 kDa) a hydrodynamic radius of 250 nm (!!!) is obtained. Comparatively, Ficoll 70 with a comparably weight (70kDa) has a hydrodynamic radius of only 5 nm. This is a factor of 50 for the same molecular weight, which is impossible.  Moreover, HA1000 (1000 kDa= 17 times Ha60) should suddenly have a much smaller hydrodynamic than HA60.  Real numbers were requested but are still missing. Without displaying real numbers (table instead of bar histogram) a comprehensive validation is impossible. This does not meet the scientific standard of this journal.

The volume fraction can physically not exceed 100% and practically not exceed more than 50% (due to gelation). However, the authors find all solutions (besides one) to be in the range of 100% or much higher than 100%. Ficoll70 has for example a volume fraction of 5%  for 10 mg/ml (Homouz et al Biophys J. 2009 Jan 21; 96(2): 671–680.)

The polydispersity is nearly constant, which might be correct, if the reference value for carrageenan  would not be wrongly determined (Webber et al . Carbohydrate Polymers 90 (2012) 1744–1749 published a value of 3.18.)

Hence, a new data set for the molecular characterization of HA is mandatory for publication. The manuscript should be rejected otherwise.

Author Response

Reviewer No 2

Comment:A molecular characterization of hyaluronic acid (HA) is crucial for a sound scientific work that attempts to address molecular crowding effects and not some kind of specific interaction. Such a characterization allows to relate physical properties, e.g. excluded volume and size to a cellular response. As long as these physical parameters are not known, all of the study is not scientifically meaningful enough for publication. It lacks basic understanding of the observed effects. Is the HA interaction dominated by excluded volume effects (macromolecular crowding) or specific interactions? Both is possible. The authors conclude already that macromolecular crowding is the driving force (title), but they cannot convince with the work as it is presented. Hence, I feel obliged to point out again, that the characterization of HA (Fig. 1) is severely WRONG. This issue should not be “hidden” in the supplemental nor addressed in a rather handwaving discussion only. Just to give examples: For HA60 (60 kDa) a hydrodynamic radiusof 250 nm (!!!) is obtained. Comparatively, Ficoll 70 with a comparably weight (70kDa) has a hydrodynamic radius of only 5 nm. This is a factor of 50 for the same molecular weight, which is impossible. Moreover, HA1000 (1000 kDa= 17 times Ha60) should suddenly have a much smaller hydrodynamic than HA60. Real numbers were requested but are still missing. Without displaying real numbers (table instead of bar histogram) a comprehensive validation is impossible. This does not meet the scientific standard of this journal. The volume fraction can physically not exceed 100% and practically not exceed more than 50% (due to gelation). However, the authors find all solutions (besides one) to be in the range of 100% or much higher than 100%. Ficoll70 has for example a volume fraction of 5% for 10 mg/ml (Homouz et al Biophys J. 2009 Jan 21; 96(2): 671–680). The polydispersity is nearly constant, which might be correct, if the reference value for carrageenan would not be wrongly determined (Webber et al . Carbohydrate Polymers 90 (2012) 1744–1749 published a value of 3.18). Hence, a new data set for the molecular characterization of HA is mandatory for publication. The manuscript should be rejected otherwise.

Response: The reviewer is correct, the data, under no circumstances, should not / must not, and as a result would not, be hidden in the supplementary information or, even worse, removed from the manuscript. After all, one of the points that we are making is that the established method to calculate % fractional volume occupancy is based on the assumption that the molecules are spherical, which is not the case in real life. We appreciate the comment though that the values should be presented in a table format and we have added a table. For the % fraction volume occupancy, we have added the following comment in the table in the results section to clearly illustrate that the values are of questionable validity: ‘In parenthesis, the calculated % fractional volume occupancy values, but as the values are > 100 %, they cannot be considered as pragmatic.’ The discussion has been re-written and in the revised version of the manuscript reads as follows: ‘With respect to % fractional volume occupancy, only the 0.5 mg/ml and 1 mg/ml HA 500 concentrations and the 0.1 mg/ml HA 1,000 concentration exhibited % fraction volume occupancy below 60 %, which is at the boundary of probability. Over the years, a diverse range of % fraction volume occupancy values have been reported in the literature using the method that we also used herein [47] [e.g. 5.2 % for dextran sulphate 500 kDa [75]; 28 % for a cocktail of dextran sulphate 10 kDa, Ficoll® 70 kDa and Ficoll® 400 kDa [51]; 9-54 % for Ficoll® 70 kDa and Ficoll® 400 kDa cocktail (subject to the concentration of each molecule in the solution) and polyvinylpyrrolidone 40 kDa and polyvinylpyrrolidone 360 kDa (subject to the concentration of each molecule in the respective solution) [32]; > 100 % for highly sulphated seaweed polysaccharides, such as carrageenan, fucoidan, galactofucan, arabinogalactan, ulvan [15]]. Although % fractional volume occupancy is frequently employed to predict the effectiveness of a macromolecule to occupy space / exclude volume in cell culture and consequently result in enhanced and accelerated ECM deposition, we believe that this approach should be treated with caution as the % fractional volume occupancy is calculated from the molecular weight and the hydrodynamic radius, based on the assumption that the molecule in question is spherical, which is not the case in real life (e.g. dextran is ribbon-like or rod-like molecule, Ficoll® is considered as a deformable sphere, as opposed to a compact sphere [76]).’ There is another comment further down in the discussion that further advocates the theoretical nature of the used / established method to determine % fractional volume occupancy that reads as follows: ‘Thus, one would have expected increasing HAs concentration, which resulted in in-creased % fractional volume occupancy, to also result in increased collagen deposition. Considering though that polydispersity was not increased as a function of increasing HAs concentration and neither did the collagen deposition, we feel that this further substantiates our aforementioned claim with respect to theoretical, as opposed to practical, nature of % fractional volume occupancy.

With respect to hydrodynamic radius and molecular weight correlation that the reviewer raised above, we would like to point out that we did not measure the hydrodynamic radius of Ficoll 70 kDa; we measured the hydrodynamic radius of the Ficoll 70 kDa and Ficoll 400 kDa cocktail. Admittedly, it was much lower than the hydrodynamic radius of HA 60 kDa. In our opinion, this can be attributed to the unmatched by Ficoll water binding capacity of HA. It is worth also noting that polydispersity index, hydrodynamic radius and fraction volume occupancy should be calculated, assessed and compared at a given concentration, optimal for the molecule studied, as high concentrations of particles result in multi-scattering or in unpredictable agglomeration, whilst too dilute samples may not generate enough light to analyse [Panchal J, Kotarek J, Marszal E, Topp EM. Analyzing subvisible particles in protein drug products: a comparison of dynamic light scattering (DLS) and resonant mass measurement (RMM). AAPS J. 2014 May;16(3):440-51; Bhattacharjee S. DLS and zeta potential - What they are and what they are not? J Control Release. 2016 Aug 10;235:337-351]. As such, we feel that detailed comparisons between the Ficoll cocktail and the HA are not valid. Considering the above, we feel that the zeta potential, hydrodynamic radius and polydispersity index (up to 0.9) values are accurate. We recognise that there is an issue with the % fractional volume occupancy, which we have clearly stated. Further, a substantial in vitro work has been conducted. Taking all these together, we are of the opinion that the manuscript is scientifically valid and suitable for publication in Cells.